

# Controls on zooplankton methane production in the central Baltic Sea

Beate Stawiarski[1], Stefan Otto[1], Volker Thiel[2], Ulf Gräwe[1], Natalie Loick-Wilde[1], Anna K.
5   Wittenborn[1,2], Stefan Schloemer[3], Janine Wäge[1], Gregor Rehder[1], Matthias Labrenz[1], Norbert
Wasmund[1], and Oliver Schmale[1]

[1]Leibniz Institute for Baltic Sea Research Warnemünde, Seestrasse 15, 18119 Rostock, Germany
[2]Geoscience Center, Georg-August University of Göttingen, Goldschmidtstr. 3, 37077 Göttingen, Germany
10   [3]Federal Institute for Geosciences and Natural Resources, Stilleweg 2, D-30655 Hannover, Germany

*Correspondence to*: Oliver Schmale (oliver.schmale@io-warnemuende.de)



## Abstract

Several methanogenic pathways in oxic surface waters were recently discovered, but their relevance in the natural environment is still unknown. Our study examines distinct methane enrichments that repeatedly occur below the thermocline during the summer months in the central Baltic Sea. In agreement with previous studies in this region, we discovered differences in the methane distributions between the Western and Eastern Gotland Basin, pointing to in situ methane production below the thermocline in the latter (conc. $CH_4$ 14.1 ±6.1 nM, $\delta^{13}C$ $CH_4$ -62.9‰). Through the use of a high resolution hydrographic model of the Baltic Sea, we showed that methane below the thermocline can be transported by upwelling events towards the sea surface thus contributing to the methane flux at the sea/air interface. To quantify zooplankton-associated methane production rates, we developed a sea-going methane stripping-oxidation line to determine methane release rates from copepods grazing on $^{14}C$-labelled phytoplankton. We found that: (1) methane production increased with the number of copepods, (2) higher methane production rates were measured in incubations with *Temora longicornis* (125 ±49 fmol methane copepod$^{-1}$ d$^{-1}$) than incubations with *Acartia* spp. (84 ±19 fmol $CH_4$ copepod$^{-1}$ d$^{-1}$) dominated zooplankton communities, and (3) methane was only produced on a *Rhodomonas* sp. diet, but not on a cyanobacteria diet. Furthermore, copepod-specific methane production rates increased with incubation time. The latter finding suggests that methanogenic substrates for water-dwelling microbes are released by cell disruption during feeding, defecation, or diffusion from fecal pellets. In the field, particularly high methane concentrations coincided with stations showing a high abundance of DMSP-rich Dinophyceae. Lipid biomarkers extracted from phytoplankton- and copepod-rich samples revealed that Dinophyceae are a major food source of the *T. longicornis* dominated zooplankton community, supporting the proposed link between copepod grazing, DMSP release, and the buildup of subthermocline methane enrichments in the central Baltic Sea.

## 1 Introduction

Climate change, caused by increased greenhouse gas concentrations in the atmosphere, has an indisputable influence on societal and economical evolution on local, regional and global scales. In order to better predict future climate development, a more precise quantitative description of individual sources and sinks of relevant greenhouse gases, such as methane, is crucial. For reliable future projections, we need to precisely understand how different environmental parameters impact source strengths, and we have to integrate the mechanistic understanding into climate change models.

Methane as an atmospheric component has a relevant impact on the earth's climate (Etminan et al., 2016; IPCC, 2013). In general, biogenic sources of methane are associated with microorganisms (Archaea) in anoxic habitats, for example, in the ocean and lake sediments, wetlands, landfills, rice fields or the gastrointestinal tracts of termites and ruminants (IPCC, 2013). However, recent studies demonstrated that methanogenesis also occurs in oxic environments. These unconventional methanogenic pathways are mediated by aerobic prokaryotes (Yao et al., 2016) as well as eukaryotes, including plants



(Keppler et al., 2006; Lenhart et al., 2016), animals (Tuboly et al., 2013), lichens (Lenhart et al., 2015), and fungi (Lenhart et al., 2012).

Methane concentrations in the oxygenated surface ocean and epi-/metalimnic lake waters show strong regional and seasonal fluctuations (e.g. Jakobs et al., 2014; Donis et al., 2017). Large areas are supersaturated with methane and act as a net source

to the atmosphere (e.g. Bange et al., 1994; Lamontagne et al., 1973; Tang et al., 2014). Thus, by contributing approximately 20 % to the global natural emissions, oceans and lakes are significant sources in the atmospheric methane budget (Bastviken et al., 2011; IPCC, 2013; Rhee et al., 2009). It is assumed that climate-driven modifications in aquatic systems such as increasing water temperatures, enhanced stratification, and nutrient limitation could further reinforce aquatic methane production (Karl et al., 2008). Such modifications are particularly important for shallow oxic methane production that

largely bypasses microbial methane consumption as it places the methane source close to the water surface, intensifying fluxes to the atmosphere. However, the origin of methane in the oxic upper water column is still unclear and has fascinated researchers during the last 40 years (Scranton and Brewer, 1977), a conundrum referred to as the "methane paradox".

Recently, a growing number of studies have identified several pathways that could explain methane enrichments in shallow aerobic waters: (I) Studies in the oligotrophic equatorial Pacific proposed that methane production in this region might be

related to the break-down of methylphosphonate (MPn), in particular under phosphate-stressed conditions (Karl et al., 2008; Teikari et al., 2018). The enzymatic cleavage of the carbon-phosphorus bond is not sensitive to oxygen and the involved C-P lyase genes are widespread across the bacterial domain – with *Pseudomonas* sp. populations as critical participants in this methanogenic pathway (Wang et al., 2017). MPn was shown to be ubiquitous in dissolved organic matter in the ocean (Repeta et al., 2016), and the MPn biosynthetic key enzyme was identified in the archaeon *Nitrosopumilus maritimus* and the

α-proteobacteria *Pelagibacter ubique*, two of the most abundant marine prokaryotes (Metcalf et al., 2012; Born et al., 2017). (II) Studies conducted at Lake Stechlin (Germany) suggest a model in which methane production is related to the activity of hydrogenotrophic methanogens attached to photoautotrophs (Grossart et al., 2011). In their model, the symbiotic community allows for anaerobic growth and a transfer of diazotroph-derived hydrogen among the microorganisms. (III) Alternatively, studies in the central Arctic Ocean suggested that methane can be produced through the degradation of

dimethylsulfoniopropionate (DMSP; Damm et al., 2010), an algal osmolyte that is abundant in marine phyto- and zooplankton (Keller et al., 1989). However, the microorganisms that mediate the metabolism of DMSP under oxic conditions are still unknown; the associated methyl reductase Mcr gene complex required for the final step of methyl reduction points to an involvement of methanogenic Archaea (Tang et al., 2016). (IV) Another recent laboratory approach showed that marine phytoplankton was also capable of producing methane (Lenhart et al., 2016). These organisms may use bicarbonate as an

unspecific carbon source in different metabolic pathways, e.g. for the synthesis of pectin and cellulose – components already known as methane precursors from terrestrial plants – where methane can be directly produced via methyl group cleavage (e.g. Keppler et al., 2008). Methionine is another substrate that was found to be transformed into methane, this substrate might be used in the algae synthesis of DMSP. (V) An older hypothesis attributed the microbial formation of methane in oxygenated surface water to anoxic microniches within decaying organic aggregates or zooplankton fecal pellets (Holmes et





al., 2000; Karl and Tilbrook, 1994; Oremland, 1979). De Angelis and Lee (1994) formulated the copepod-gut flora theory based on their observations of methane production during experimental zooplankton grazing on unialgal cultures. The authors used selected growth stages (adult and C5 stages) from pure cultures of *Temora longicornis*, *Acartia tonsa*, and *Calanus pacificus*, which had been isolated from the coastal waters from the North Atlantic. In these laboratory experiments,
methane production was only detected in incubations containing *T. longicornis*.

The Baltic Sea is a nontidal semi-enclosed brackish basin connected to the North Sea (Fig. 1). This estuarine system is characterized by pronounced lateral gradients including e.g. temperature, salinity, oxygen and methane concentrations, which altogether strongly influence the biology of the ecosystem (Feistel et al., 2008 and references therein). The Baltic Sea consists of several sub-basins, of which the Eastern Gotland Basin located in the central Baltic Sea is the largest. The Eastern
Gotland Basin has a maximum depth of 248 m and establishes a seasonal thermocline at 10–30 m water depth in the summer months, and a permanent halocline at 70-90 m (Omstedt et al., 2004). In general, bottom waters in this basin are anoxic, unless, episodic inflow events of more saline and oxygen rich North Sea water into the deep basins lead to temporary oxic conditions (Franck et al., 1987; Mohrholz et al., 2015).

Methane concentrations in the Baltic Sea tend to increase with depth due to its release from the anoxic sediments (Schmale
et al., 2010). However, recent studies in the central Baltic Sea revealed a recurring methane accumulation in oxic waters immediately below the thermocline during the summer months (Jakobs et al., 2014; Schmale et al., 2018). Stable carbon isotopes indicated an in situ biogenic methane origin, whereas clonal sequences pointed towards methanogenic Archaea as potential producers (Schmale et al., 2018). It was further shown that zooplankton-associated methane production contributed to the subthermocline methane enrichment (Schmale et al., 2018). The authors suggested that changes in the copepod
community and food web structure influenced the spatial heterogeneity of methane accumulation in the upper part of the water column. In the same study, Schmale et al. (2018) investigated natural bulk zooplankton communities obtained from the depth of the subthermocline methane enrichment, and reported an increase of methane with increasing amount of zooplankton. A mass balance established for the oxic part of the water column indicated that zooplankton-associated methane production rates alone were not sufficient to fully explain the observed methane enrichment. However, the
production rates were obtained from incubations with very dense and probably food limited zooplankton communities (1000 times the natural density), leading to results that are difficult to transfer into the natural environment (Schmale et al., 2018). To overcome these experimental deficits, we developed a methane stripping-oxidation line to study zooplankton-associated methane production rates in the field. Our ship-based incubation experiments were performed with different zooplankton communities obtained from either the surface or from below the thermocline. We used almost natural copepod-densities
(1.5–8.5 times the natural density), and different diets of $^{14}$C-labeled phytoplankton. In conjunction with field investigations of methane concentrations and plankton distributions, the present study gives important insights into the controls of zooplankton-associated methane production in the central Baltic Sea.



## 2 Methods

### 2.1. Hydrographical and chemical characteristics of the water column

During the cruise AL483 on R/V *Alkor* in August 2016 in the central Baltic Sea, seven stations in the Western (TF0283, TF0284) and the Eastern (from south to north: TF0250, TF0260, TF0271, TF0286, TF0285) Gotland Basin (Fig. 1) were

5    investigated. At each station, the hydrographical variables, including temperature, salinity and oxygen concentrations, of the water column were examined along vertical profiles using a SBE 911 plus CTD system (Seabird Electronics, USA, see supplement S1). In addition, the vertical methane distribution pattern and the stable carbon isotope ratio of methane ($\delta^{13}$C CH$_4$) were measured to differentiate between regions affected and regions unaffected by subthermocline methane production. For these studies, subsamples were taken from the rosette water sampler and analyzed in the home laboratory

10    using a purge and trap system for methane concentration measurements and a continuous flow Isotope Ratio Mass Spectrometer IRMS for stable carbon isotope analyzes (see supplement S1). In order to investigate whether station TF0284 was affected by coastal upwelling during the time of sampling, we analyzed the output of a numerical ocean model, covering the North Sea / Baltic Sea. This hydrodynamical model computed a reconstruction of the state of the Baltic Sea with a spatial resolution of 1 nautical mile and 50 vertical levels. Further details of the model are given in Gräwe et al. (2015).

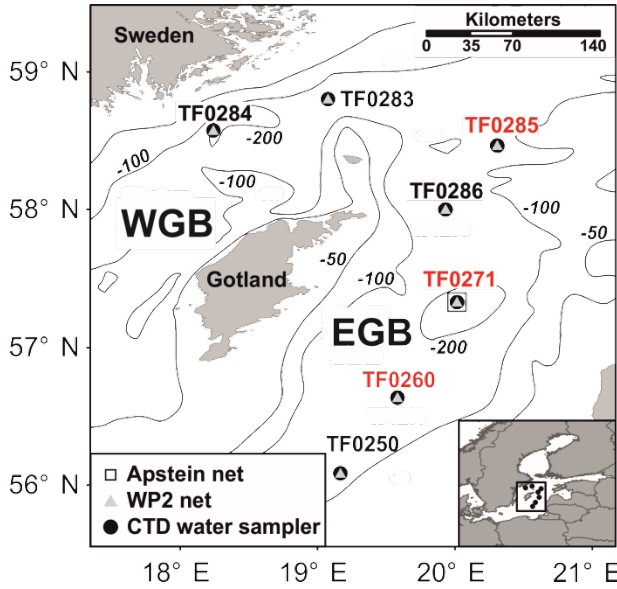

**Figure 1: Map of sampling stations in the Western Gotland Basin (WGB) and the Eastern Gotland Basin (EGB). Symbols describe the sampling gear. Stations with a distinct subthermocline methane enrichment are marked in red.**



## 2.2 Plankton community and lipid biomarker analyses

Phytoplankton samples for water column community analysis were taken from the 10 L free-flow bottles at stations TF0271, TF0284 and TF0286 (Fig. 1). Equal amounts of water from the mixed layer (depths: 0–1 m, 2.5 m, 5 m, 7.5 m and 10 m) were pooled in accordance with the guidelines of the Helsinki Commission (HELCOM, 2017). At stations characterized by a

distinct subthermocline methane enrichment (TF0271, TF0260, TF0285 and TF0286), additional samples were taken from the subthermocline chlorophyll *a* maximum at about 20 m water depth. At station TF0271, three samples were taken on August 12[th] and 18[th] to investigate the temporal variability in the community composition at this location. All phytoplankton samples were transferred into 250 ml brown glass bottles and preserved with 1 ml of acetic Lugol solution (2 % final concentration). For later community analyses, 25 ml sub-samples were concentrated in settling chambers (Utermöhl, 1958).

They were counted using an inverted microscope and the counts were converted into carbon biomass using the cell volumes (Olenina et al., 2006; HELCOM (2017)).

Zooplankton samples for water column community analyses were collected with a WP2 net (towed at 0.5 m s$^{-1}$, mouth opening, 25 m$^2$, mesh size 100 μm, according to HELCOM (2012)) independent of daytime. Vertically integrated hauls from two depth intervals were taken to obtain zooplankton samples: (i) thermocline to surface (e.g. 20-0 m, depending on the

physical structure of the water column), and (ii) halocline to thermocline (e.g. 60-20 m). The concentrated samples (500 ml) were preserved in borax-buffered formalin (4 % final concentration). For later copepod-specific community analyses, sub-samples were counted using a compound microscope until at least 500 individual copepods were taxonomically classified. Nauplii were pooled together for all copepod species, while C1-C5 copepodite stages and adults were pooled for individual species. Finally, to exclude potential daytime effects through vertical migration of the zooplankton communities above the

halocline, counts from both integrated hauls were averaged according to the filtered volume. Similar to the phytoplankton sampling strategy, three samples at TF0271 were taken in a period of ten days to investigate the temporal variability in the community composition at this station.

Lipid biomarkers were analyzed to obtain information on the trophic relationships of copepods in the field at station TF0271 that showed a distinct methane enrichment below the thermocline. For this purpose, their putative food source,

phytoplankton in the mixed layer, was sampled with an Apstein-net. The phytoplankton was separated from co-sampled zooplankton using a simple self-built trap consisting of a 1.5L transparent plastic bottle with a closable outlet at the bottom. The zooplankton was attracted towards the trap using a light source attached close to the outlet, and drained off. Then the phytoplankton, which remained in the bottle, was sampled. The target copepod *T. longicornis* has been known to migrate diurnally from the light-penetrated surface layer towards greater depths to escape predation (Hansen et al., 2006; Schmale et

al., 2018). For lipid biomarker studies, a concentrated sample of zooplankton rich in *T. longicornis* was retrieved, avoiding major co-sampling of phytoplankton. This sampling was performed at station TF0271 by hauling a WP-2 net (see above) in the subthermocline layer (25-60 m) during the daytime. For comparison, a zooplankton sample was taken at station TF0250 without a distinct methane enrichment below the thermocline. Here, the community composition was low in *T. longicornis*



and instead was dominated by *Acartia* spp. and *Pseudocalanus* spp. All samples were concentrated by sieving (20 µm) and kept frozen until further workup in the home laboratory. The samples were lyophilized and extracted (3x, ultrasound, 15 min, 20°C) with dichloromethane (DCM)/methanol (MeOH) (2:1, v:v), DCM/MeOH (3:1, v:v) and DCM/*n*-hexane (2:1, v:v). From the resulting total organic extract, neutral lipids (NL, largely containing storage lipids such as triglycerides, wax esters, and sterols) were separated using silica gel column chromatography by elution with DCM/acetone (9:1, v:v). After drying, fatty acids (FA) in the NL fraction were transesterified by reaction with trimethylchlorosilane (TMCS)/MeOH (1:9, v:v; 90 min, 80 °C). After partitioning into *n*-hexane (3x, 1 mL) and drying, alcohols contained in the NL were converted to their trimethylsilyl (TMS-) derivatives by reacting with 200 µl of a *n*-hexane/BSTFA/pyridine mixture (5:3:2, v:v:v; 40 °C, 60 min). The derivatized NL fractions were analyzed by coupled gas chromatography – mass spectrometry (GC-MS), as described elsewhere (Thiel and Hoppert 2018).

## 2.3 Sampling for ship-based laboratory experiments

Three ship-based grazing experiments were conducted at station TF0271, where a persistent and distinct methane enrichment below the thermocline was detected during the cruise. These experiments were designed to examine how (i) the abundance of copepods and (ii) their food source impacts zooplankton-associated methane production, and how (iii) the methane production rates vary between different copepod communities. To measure copepod species-specific methane production rates, zooplankton communities were sampled from the surface- (e.g. 20-0 m) and from the subthermocline waters (e.g. 60-20 m). In the Central Baltic Sea, these layers are commonly dominated by *Acartia* spp. and *T. longicornis*, respectively (Hansen et al., 2006) and will be referred to as the surface and the subthermocline communities. For the sampling hauls the cod end of the WP2 net was sealed from outside and towed 0.1 m s$^{-1}$ to reduce damage to the zooplankton, and the content was transferred immediately into a 25 L bucket filled with seawater from the depth where the zooplankton was sampled (i.e. surface or subthermocline water). The zooplankton mainly comprised of copepods and was left in the cold room at 10°C for an hour to allow damaged individuals to settle to the bottom of the bucket. A subsample of living copepods was removed gently with a 500 ml beaker from the upper layer of the bucket to avoid injured animals, and checked for species composition under the dissecting microscope before subsamples were used for the grazing experiments. The sampling took place at 14:00 UTC, before vertical migration of one of the target copepods, *T. longicornis*, started (Hansen et al., 2006; Schmale et al., 2018). This time was chosen because it was more likely that *T. longicornis* in the deeper water column would be starved and thus, would start grazing within the experiments.



**Table 1: Experimental conditions for the zooplankton grazing experiments (exp.). SA$_{phy}$ is the specific activity of the phytoplankton fed to the copepods. POC stands for particulate organic carbon. No. of copepods is the average amount of copepods used in the incubation experiments.**

| Exp. | Food | SA$_{phy}$ (MBq mmol$^{-1}$) | POC conc. at start (mg C L$^{-1}$) | POC conc. at end (mg C L$^{-1}$) | Zooplankton community | No. of copepods | Duration of the exp. (d) |
|---|---|---|---|---|---|---|---|
| 1 | *Rhodomonas* sp. | 695.23 | 1.34 ± 0.04 | 1.24 ±0.08 | sub-thermocline | 25 ±3<br>56 ±2<br>73 ±2 | 1 |
| | | | 1.22 ± 0.04 | 3.57 ±0.38 | surface | 25 ±8<br>50 ±12<br>76 ±18 | |
| 2 | *Rhodomonas* sp. | 613.46 | 1.61 ± 0.61 | 1.73 ±0.45 | sub-thermocline | 64 ±3 | 1-3 |
| 3 | *Nodularia spumigena* | 299.33 | 3.17 ± 0.26 | 3.09 ±1.18 | sub-thermocline | 66 ±7 | 1-3 |

Two phytoplankton compositions were used to test for the influence of the diet on zooplankton-associated methane
production: (i) a *Rhodomonas* sp. (Cryptophyceae) laboratory culture, and (ii) a *N. spumigena* (Cyanophyceae) dominated
culture from the surface mixed layer in the field, which is typically found in these waters of the central Baltic Sea in summer
(Wasmund, 1997). The laboratory culture of *Rhodomonas* sp. was grown in f/2 medium (Guillard, 1962), prepared in
autoclaved bottles from 0.2 µm filtered seawater from 10 m depth. A new sub-culture was established every 3-5 days and the
inoculum was chosen to be sufficient to keep the culture in exponential growth (cf. Knuckey et al., 2005). The second
phytoplankton culture was established from the top chlorophyll *a* fluorescence maximum (0-10 m) sampled with an Apstein
net (55 µm mesh size, Hydro-Bios). The culture was re-suspended in 500 ml of seawater and left for an hour to allow the *N.
spumigena* cells to float on top. The cells were carefully collected with a pipette and transferred into an autoclaved 1 L glass
bottle, which was filled up with surface seawater. Then, 36.2 µM of NaH$_2$PO$_4$ as for the f/2 medium were added. Both the
*Rhodomonas* sp. and the *N. spumigena* culture bottles were incubated in 90 L tubs, which were placed in a shaded location
on deck and continuously flushed with sea surface water at a temperature of ~18.5°C. A third phytoplankton culture was
collected from a subthermocline chlorophyll *a* peak closely below the thermocline (~20 m depth), where mixotrophic
Dinophyceae were expected to be dominant (Carpenter et al., 1995; Hällfors et al., 2011). For this, up to 150 L of seawater
were obtained from a CTD rosette and concentrated on a 20 µm mesh. However, this culture did not grow sufficiently under
similar temperature and light conditions as in situ, and could not be used within the grazing experiments.



### 2.4 Phytoplankton [14]C-labelling

Once a phytoplankton culture was selected for grazing experiments, three aliquots were split between three autoclaved 1 L incubation bottles (Duran) and filled up with the corresponding medium. One of these bottles was spiked with 18.5 MBq of [14]C-labelled sodium bicarbonate (NaHCO$_3$, 2.18 GBq mmol$^{-1}$, Moravek Biochemicals, USA) for [14]CH$_4$ production measurements (Sect. 2.5.1 Methane production and consumption rates). The other two bottles received equal amounts of unlabeled NaHCO$_3$ for measuring the particulate organic carbon concentration (Sect. 2.5.2 Analysis of particulate organic carbon) and microbial methane consumption (Sect. 2.5.1 Methane production and consumption rates). All bottles were incubated for 3-5 days to allow cells to grow and to take up the [14]C-label (Welschmeyer & Lorenzen, 1984). The specific activity of the labeled phytoplankton (SA$_{phy}$) was measured daily by filtering 1-5 ml of the culture through a 0.45 µm cellulose nitrate filter (Millipore) and rinsing it thoroughly with Milli-Q water. The filter was then dissolved in Filter Count Scintillation Cocktail (Perkin Elmer) by vortexing and the amount of [14]C-label that was incorporated into particles was measured on a liquid scintillation counter (Perkin Elmer, Tri-Carb 2800TR). Blanks for the specific activity of the cultures were collected and analyzed immediately after the label had been added to the incubation bottle.

The specific activity of phytoplankton was calculated using Eq. (1). It was used to convert the measured disintegrations per minute (dpm) obtained from the seagoing methane stripping-oxidation line (Sect. 2.5.1) to units of mmol [14]CH$_4$ produced.

$$SA_{Phy}\left[\frac{MBq}{mmol}\right] = \frac{Desintegrations_{filter}\left[\frac{dpm}{ml}\right] \times Volume_{culture}\ [ml]}{5.95 \times 10^7 \left[\frac{dpm}{MBq}\right]} \times \frac{1}{activty_{added}\ [MBq]} \times SA_{tracer}\left[\frac{MBq}{mmol}\right] \qquad (1),$$

where $SA_{Phy}$ is the specific activity of the phytoplankton, $Disintegrations_{filter}$ are the disintegrations ml$^{-1}$ on the filter, $Volume_{culture}$ is the total volume of the incubated culture, 5.95 x 10$^7$ dpm MBq$^{-1}$ is the constant for converting dpm into units of MBq, $activity_{added}$ is the total acitivity of the tracer, which was added to the incubation and $SA_{tracer}$ is the specific activity of the tracer.

### 2.5 Zooplankton grazing experiments

Three experiments with zooplankton grazing on phytoplankton (Table 1) were conducted. In experiment 1 we tested whether there was (i) a linear relationship between methane produced and the number of copepods incubated, and (ii) a difference in methane production between the surface and subthermocline zooplankton communities. In experiment 2 only the subthermocline zooplankton community was used and the incubation time was varied from 1 to 3 days to test if the increase in methane was stable over time and the production rate per copepod stayed constant. In experiment 1 and 2 a laboratory strain of *Rhodomonas* sp. (Cryptophyceae), a standard food for copepod culturing (Dutz et al., 2008) was fed to the zooplankton communities since it represents. *Rhodomonas* sp. may be considered a model representative of the Cryptophyceae, which only contributed a minor percentage to the total phytoplankton biomass in the Baltic Sea (Fig.8 and 9;





N. Wasmund, personal communication). In experiment 3 the subthermocline zooplankton community was fed the cyanobacterium *N. spumigena* and the incubation time was varied the same way as in experiment 2. Here we selected *N. spumigena* as a food source, because this species was the dominant phytoplankton in the surface waters during our field campaign. The different phytoplankton communities fed in these experiments allow one to assess the impact of the food

source on zooplankton-associated methane production rates.

Each of the 3 grazing experiments consisted of 3 sets of incubations. The first set of incubations was conducted with zooplankton grazing on [14]C-labelled phytoplankton for methane production measurements (Sect. 2.5.1 Methane production and consumption). The second set of incubations was conducted with zooplankton grazing on unlabeled phytoplankton to determine the food availability over the course of the grazing experiments (Sect. 2.5.2 Analysis of particulate organic

carbon). Also the third set of incubations was conducted with zooplankton grazing on unlabeled phytoplankton to measure the loss of methane by microbial oxidation under the experimental conditions (Sect. 2.5.1 Methane production and consumption).

In selective incubations the oxygen saturation was monitored over time using oxygen spots (5 mm, PreSens Precision Sensing) and an optode (Single Channel Oxygen Meter, Fibox 3 LCD, PreSens Precision Sensing). The average oxygen

saturation was 75.2 ±2.9 % throughout the experiments.

### 2.5.1 Methane production and consumption rates

For the methane production incubations, 250 ml gastight bottles (Duran) were used. The bottle lids were modified with one inlet and one outlet tube (Fig. 2), each sealed with autoclaved silicone stoppers. The inlet tube was long enough to reach into the lower third of the medium and had a glass frit attached at the end. The outlet tube was short enough to remain in the

headspace of the bottle. Each bottle was filled with 50 ml of [14]C-labelled phytoplankton, 3-15 ml of zooplankton stock culture and topped up to a total volume of 200 ml with 0.2 µm filtered seawater from either 10 m or 20 m depth, according to the depth from which the zooplankton community was obtained. All bottles were kept under in situ temperature for 1 to 3 days in temperature-controlled incubators. The methane, which was produced within the time of incubation, was finally measured with a methane stripping-oxidation line (modified after de Angelis and Lee, 1994; Fig. 2; details to the work

principle in supplement S1). In brief, the bottles were connected to the line, the water samples purged with helium carrier gas and the methane (and other hydrocarbons) was concentrated on a cold trap. After heating the trap the methane was released and separated from other hydrocarbons by gas chromatography (GC) and transferred to a furnace where it was converted to [14]C-labelled carbon dioxide ($CO_2$). Finally, the [14]$CO_2$ was trapped and the activity measured by liquid scintillation counting.

Blank incubations were conducted with phytoplankton and seawater only and the volume of zooplankton was replaced with

the corresponding amount of GF/F (Whatman) filtered seawater in which the zooplankton was kept before incubations. The $CH_4$ concentrations of all blank incubations were not different from the blank of the methane stripping-oxidation line (Mann-Whitney U test, p = 0.9, df = 1).





Microbial methane oxidation rates were measured and the production rates corrected accordingly. For these measurements, [14]C-labelled methane was injected into 600 ml incubation bottles, which were filled with the corresponding volumes of plankton and water as in the incubations for methane production measurements. The amount of produced $^{14}CO_2$ was measured according to the method of Jakobs et al. (2013). The rates were similar in all incubations and in the range of 0.13-

0.44 pM d$^{-1}$. Neither the presence of copepods, nor the composition of their community or their food had a significant influence on the methane consumption rates. In incubations with *Rhodomonas* sp., methane oxidation rates were 0.13-0.39 pM d$^{-1}$ and accounted for a loss of 0.2-2.3 % of the produced methane. In incubations with cyanobacteria, the oxidation rates were slightly higher (0.3-0.44 pM d$^{-1}$).

All sets of incubations included 2-3 replicates. For estimating the linear or exponential trends in methane concentration or

production in relationship with the number of copepods or time, the Gauss-Newton method was used for minimizing the sum of squares between the fits and the measurements (Mystat 12, Systat software).

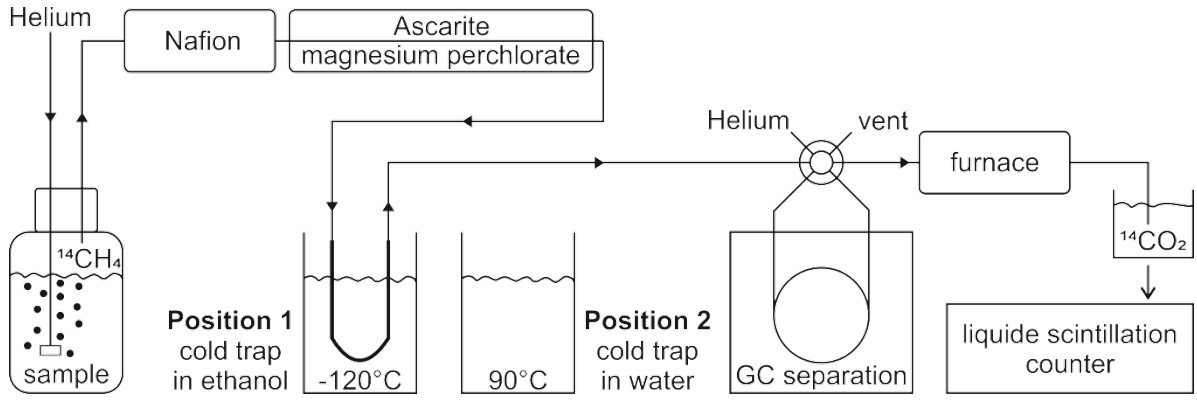

**Figure 2: Schematic view of the methane stripping-oxidation line. Position 1 (displayed): the cold trap is placed in a -120°C ethanol bath to retain the hydrocarbons. Position 2: the cold trap is transferred into a water bath at 90°C to release the hydrocarbons towards the gas chromatograph (GC). A detailed description is included in supplement S1.**

### 2.5.2 Analysis of particulate organic carbon

The particulate organic carbon (POC) content in the incubations was quantified at the beginning and end of our incubation experiments to determine the food availability over the course of the grazing experiments. For this, a larger volume of sample was required for later POC analysis using a Carlo Erba Elemtal Analyser (Typ EA 1110; Carlo Erba; Nieuwenhuize

et al., 1993), and 1 L bottles (Duran) were used for incubations. To achieve similar concentrations of zooplankton and phytoplankton (ml$^{-1}$) as in the first set of incubations, stock culture volumes were increased accordingly. For ambient POC analysis, one 50 ml sample was taken from each bottle at the beginning and at the end of the experiment on pre-combusted GF/F filters. The initial sample was removed from the experimental bottle before the copepods were added to it, and the final sample was taken after copepods had been removed with a 50 µm sieve. All POC measurements (Table 1) at the beginning

and at the end of the incubations exceeded the threshold for food limitation of 1-0.5 mg C L$^{-1}$ (Berggreen et al., 1988). As the



food was diluted with 0.2 µm filtered seawater, all POC at the beginning of the experiments can be assigned to the diet that was added. However, at the end of the grazing experiments, excreted fecal pellets of the zooplankton may have added up to the organic carbon pool. Further, it was difficult to separate the copepods from *N. spumigena* by the 50 µm sieve, as *N. spumigena* typically exceeds 50 µm in size. In consequence, no grazing rates based on the decrease in POC were calculated

for the individual incubations.

## 3 Results and Discussion

### 3.1 Subthermocline methane – distribution, sources, and sea/air exchange

Surface waters were oversaturated with methane with respect to the atmosphere at all stations (saturation values between 118

and 198 %) using the mean atmospheric methane concentrations for August 2016 obtained from atmospheric tower measurements at Utö (1919 ppb; position 59° 46'50N, 21° 22'23E; Finnish Meterological Institute, J. Hatakka, personal communication). However, the Eastern Gotland Basin had much higher maximum subthermocline methane concentrations than the Western Gotland Basin ($14.1 \pm 6.1$ nM vs. $8.7 \pm 0.3$ nM; Fig. 3). Such regional differences have previously been documented in the central Baltic Sea (Schmale et al., 2010; Jakobs et al., 2014; Schmale et al., 2018).

Concentration profiles and stable carbon isotopes obtained in these earlier studies indicated that the subthermocline methane enrichments resulted from in situ production within the oxic water body. A transport from the deep anoxic waters was excluded because the methane from this pool is efficiently oxidized by aerobic methanotrophic bacteria situated in the oxic/anoxic transition zone at about 100 m water depth (Jakobs et al., 2013; Schmale et al., 2012; 2016). As their metabolism favors the turnover of $^{12}CH_4$, the remaining $^{13}CH_4$ becomes enriched and $\delta^{13}C$ $CH_4$ values in the respective water layer are

comparably high (e.g. -40‰ in 80 m water depth; Jakobs et al., 2013). In our current study, subthemocline methane enrichments in the Eastern Gotland Basin were characterized by strikingly depleted $\delta^{13}C$ $CH_4$ values (-62.9‰ at 27 m at TF0271, Fig. 3), supporting the idea that the pronounced methane anomaly in this area originated from in situ biogenic production. In contrast, stable isotope ratios of methane in the upper water column of the Western Gotland Basin showed $\delta^{13}C$ $CH_4$ values of -47.7‰ at 20 m water depth (TF0284, Fig. 3) that are close to atmospheric equilibrium (-47‰).




**Figure 3: Vertical profiles of temperature (T), oxygen (O₂) and methane concentrations (CH₄) in the central Baltic Sea: (a) stations with a distinct, and (b) stations without a distinct subthermocline methane enrichment; stable carbon isotope values of methane (δ¹³C) are presented for station TF0284 (Western Gotland Basin) and TF0271 (Eastern Gotland Basin).**

5   Seasonal observations in the Baltic Sea revealed that the development of a thermocline, which functions as a barrier and limits fluxes to the atmosphere, was essential for the build-up of subthermocline methane enrichments (Jakobs et al., 2014; Schmale et al., 2018). Upwelling events can offset water column stratification through a replacement of warm, mostly nutrient-depleted surface water by cooler and usually nutrient-enriched subthermocline waters (Gidhagen, 1987; Lehmann and Myrberg, 2008; Reissmann et al., 2009). Such events can significantly increase surface water methane concentrations in

10   the area around Gotland during the summer (Gülzow et al., 2013; Schneider et al., 2014). During our field campaign the sea surface temperature at station TF0284 in the Western Gotland Basin dropped from 18°C to 12°C, as indicated by our oceanographic model output (Fig. 4, see also the temperature profile of station TF0284 in Fig. 3). Assuming the thermocline was located at 18-20 m depth we estimated that the upwelled water masses must have had originated from a depth of 25-35



m. Upwelling of cold, methane-rich subthermocline water can plausibly explain the strong surface water methane oversaturation observed at TF0284 (saturation value of 198 %). In contrast to the deep water methane pool that is efficiently separated from the surface water through the halocline in about 60 m depth (Schmale et al., 2010; Jakobs et al., 2014) upwelling of subthermocline waters has to be considered as an important mechanism that contributes to the sea/air methane

5    fluxes in the Baltic Sea.

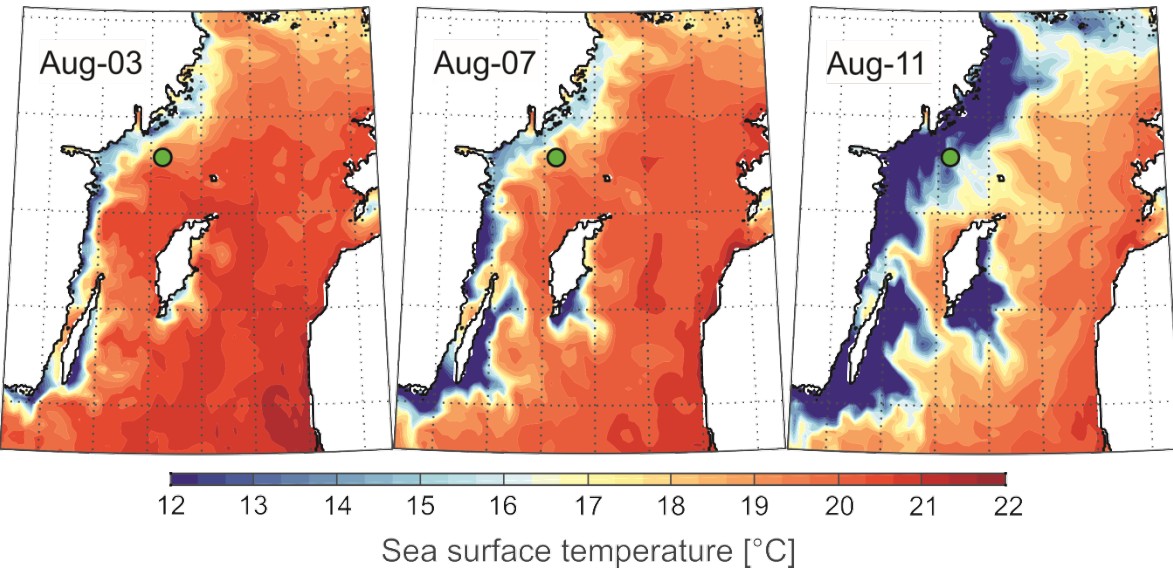

**Figure 4: Development of sea surface temperatures in the central Baltic Sea between August 3$^{rd}$ and August 11$^{th}$ 2016 using the oceanographic model of Gräwe et al. (2015). Green dot marks station TF0284.**

**3.2 Controls on zooplankton associated methane production**

10   Using the methane stripping-oxidation line we obtained species- and food-specific methane production rates in incubations, which contained field copepods (surface and subthermocline communities, Fig. 5) in nearly natural abundances (1.5-8.5 times the natural density). We found a positive correlation between methane production and the number of copepods in the incubations, but no production in controls containing phytoplankton only (Fig. 6). This implies that the production of methane is associated with the active grazing of zooplankton.





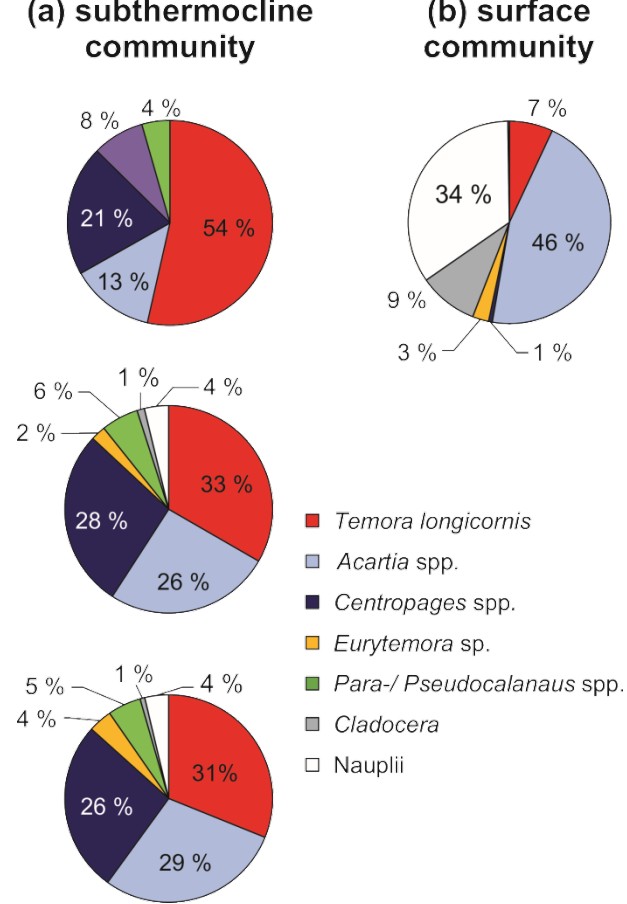

**Figure 5: Relative zooplankton community composition of the three zooplankton grazing experiments listed in Table 1. (a) subthermocline (used in experiment 1-3), and (b) surface zooplankton community composition (used in experiment 1). Adults and copepodite stages C1-C5 were pooled for individual species, but Nauplii were pooled together for all copepod species.**

The incubations with a high proportion of *T. longicornis* had higher production rates than the *Acartia* spp. dominated setups (125 ±49 vs. 84 ±19 fmol $CH_4$ copepod$^{-1}$ d$^{-1}$). This indicates that methane production depended on the composition of the zooplankton community (Fig. 6a). Similar observations were made in laboratory experiments using cultured species from the North Atlantic (De Angelis & Lee, 1994). However, these authors observed methane production in all experiments for *T. longicornis* grazing on phytoplankton, but not for *Acartia tonsa*. These differences may be due to species-specific differences in grazing rates, food preferences and gut floras. Furthermore, the methane production rates per copepod reported by de Angelis & Lee (1994) were two orders of magnitude higher (4-20 pmol $CH_4$ copepod$^{-1}$ d$^{-1}$) than the rates measured in our experiments. Still, our results are in agreement with those of previous zooplankton incubation experiments conducted in the central Baltic Sea (0.3 pmol $CH_4$ copepod$^{-1}$ d$^{-1}$ (Schmale et al., 2018)). This similarity is notable as the previous experiments used zooplankton abundances that were about 1000 times higher than the natural density in the field. The obvious discrepancy from the methane production rates reported by de Angelis & Lee (1994) might be related to the




physiological differences (e.g. animal size) between the copepods used (length *T. longicornis* North Atlantic: 1300 μm vs. Baltic Sea: 700 μm). Animal size affects the oxygen gradient in the guts of these copepods and affects the dimension of produced fecal pellets and thus, the penetration depth of oxygen into the pellet (Ploug and Jörgensen, 1999; Tang et al., 2011). Alternatively, lower methane production rates observed in our experiments may have reflected a response of the

5 animals to stress of being removed from their natural environment, kept in incubation bottles and also of being fed phytoplankton that is not representative for the phytoplankton biomass of the Baltic Sea (i.e. the cryptophyte *Rhodomonas* sp.) and thus does not belong to their natural food source. The rates may also be lower for younger development stages of copepods, which contributed to the natural surface community used in our incubations (Fig. 5).

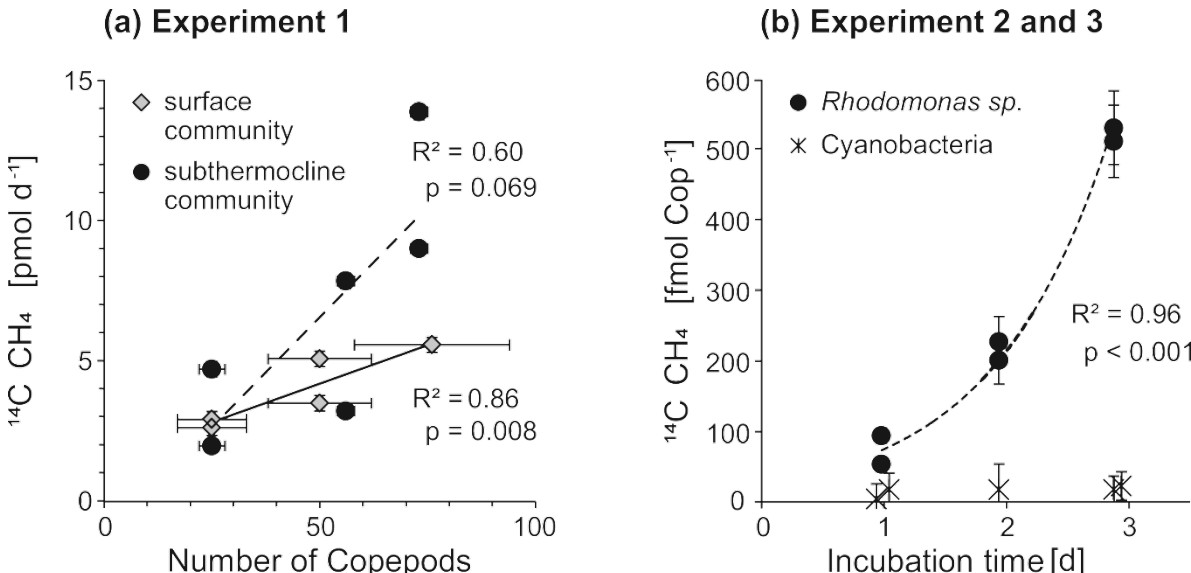

**Figure 6: (a) Experiment 1: Methane production as a function of the number of copepods in the surface (dominated by *Acartia* spp.) and subthermocline zooplankton communities (dominated by *Temora longicornis*). (b) Experiment 2 and 3: Copepod-specific methane production over time when using the cryptophyte alga *Rhodomonas* sp. (experiment 2) or the cyanobacterium *Nodularia spumigena* (experiment 3) as food sources.**

Measurable methane production occurred when the *T. longicornis* dominated community fed on *Rhodomonas* sp., while no

production occurred when it was fed on the cyanobacterium *N. spumigena* (Fig. 6b). In fact it appears that these cyanobacteria are a rather negligible food source for planktonic herbivores due to their toxic properties, large size, and low lipide concentrations (Sellner et al., 1994, Eglite et al, 2018). Our experiments therefore reveal that the availability of a suitable phytoplankton diet is an important control on the zooplankton-associated methane production.



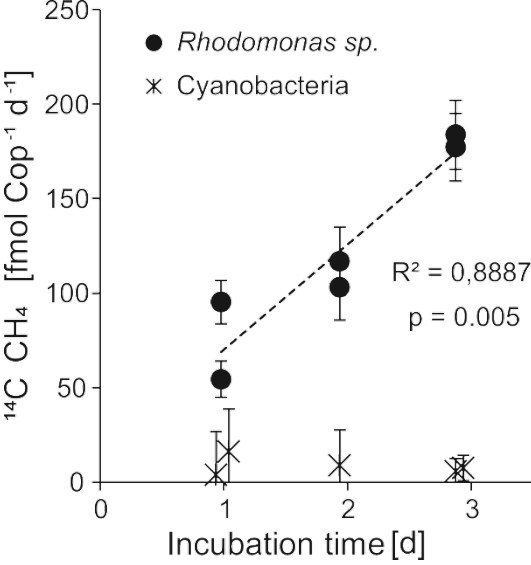

**Figure 7: Experiment 2 and 3: Copepod-specific methane production rates over time for the two food sources *Rhodomonas* sp. and cyanobacterium *Nodularia spumigena*.**

The observed exponential increase of methane with increasing incubation time implies that zooplankton-associated methane

production is a continuous process (Fig. 6b). In addition, we measured a linear increase of the methane production rates per individual with increasing incubation time (Fig. 7). Potential explanations include (1) a delay in grazing by stress through experimental conditions, (2) the accumulation of fecal pellets within the incubation bottles followed by methane production in anoxic microenvironments, and (3) enhanced methane production through release of methane precursor substances from fecal pellets or from disrupted phytoplankton cells into the incubation water. In the first case we would expect the production

rates to stabilize within the three days of the experiment, instead we observed a linear increase and a rather low variability among the replicates. In the second case, the fecal pellets could temporarily act as anoxic microenvironments for methanogenic archaea (Oremland et al., 1979; Bianchi et al., 1992; Marty et al., 1993; Karl & Tilbrook, 1994; Ditchfield et al., 2012). However, it is debatable whether anoxic conditions can persist within fecal pellets outside of the anoxic digestive tracks of the copepods. For *T. longicornis* fed on *Rhodomonas* sp., the diffusive boundary layer of the fecal pellets through

which the exchange of gases occurs was shown to be very thin, and no indications of anoxic conditions were detected (Ploug et al., 2008). Studies investigating the anoxic potential within particle aggregates were only able to confirm anoxic conditions in the interior of nutrient and carbon rich particles >600 µm and suggest that anoxia in marine aggregates is more likely to occur in an oxygen-depleted water column (Ploug et al., 1997; Ploug et al., 2001). On our cruise, a fecal pellet size of only <150 µm was measured for the surface and subthermocline zooplankton communities. We therefore believe that

anaerobic methanogenesis by archaea thriving within fecal pellets played only a minor role. Rather, we suggest that the continuous release of methanogenic substrates (like DMSP) by cell disruption during feeding, defecation, or diffusion from




fecal pellets resulted in an enrichment of these substances in the incubation water, and fostered a subsequent microbial turnover of these precursors outside the body of the copepod.

### 3.3 DMSP as a possible substrate for methane production in oxic waters

The phytoplankton community composition in the surface (Fig. 8) and subthermocline waters (Fig. 9) was similar at all

investigated stations during our field campaign. However, Dinophyceae, in particular the mixotrophic *Dinophysis norvegica*, were more abundant at stations with a distinct subthermocline methane enrichment. Dinophyceae produce relatively high amounts of DMSP as compared to the other phytoplankton species observed within our study (Keller et al. 1989, Caruana & Malin 2014). Damm et al. (2010) suggested that the microbial metabolization of DMSP and its degradation products dimethylsulfide (DMS) and methanethiol to methane is favored under nitrogen-stressed conditions in oligotrophic waters. In

the central Baltic Sea the dissolved inorganic nitrogen pool that builds up over the winter months is already exhausted after the first spring bloom in the photic zone and leaves behind nitrogen-stressed conditions for phytoplankton taxa, which are unable to fix molecular nitrogen (Schneider et al., 2009). These nitrogen-stressed conditions can lead to an accumulation of DMSP in phytoplankton cells (Sunda et al., 2007) that exceeds 10 % of the cell carbon (Matrai, 1994). In the central Baltic Sea a pronounced DMS maximum was detected during summer months in the surface waters (Leck et al., 1990). No

correlation was identified between high DMS concentrations and any particular phytoplankton species. Instead, DMS production in the surface water was accelerated through phytoplankton growth under nitrogen-limited conditions, and correlates significantly with copepod and total zooplankton biomass. Hence, the release of DMSP and DMS from phytoplankton was suggested to be controlled by zooplankton and heterotrophic bacteria (Leck et al., 1990; Kwint et al., 1996; Wolfe et al., 1997; Simo et al., 2002).





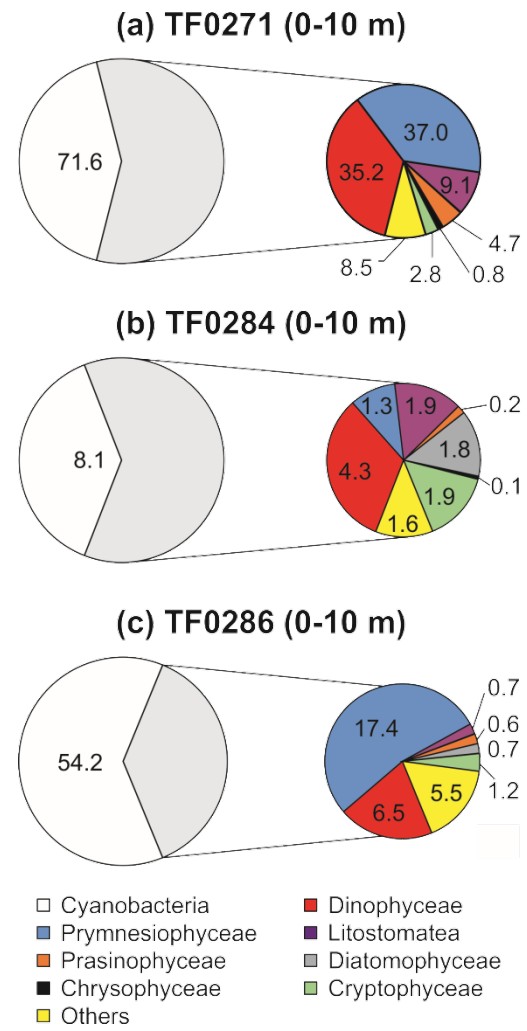

**Figure 8: Surface phytoplankton community composition integrated over the upper 10 m of the water column (mg C m$^{-3}$) at stations with a distinct (a) and without a distinct subthermocline methane enrichment (b and c). The gray shaded non-cyanobacteria community is described in detail in the colored circles.**



**Figure 9: Phytoplankton community composition within the subthermocline chlorophyll *a* maximum (mg C m⁻³) at three stations with a distinct subthermocline methane enrichment (a-e) and at one station without a distinct enrichment (f). The gray shaded non-cyanobacteria community is described in detail in the colored circles. Three replicate samples were taken at TF0271 in a period of seven days (a-c) showing that the prevalence of Dinophyceae below the thermocline is a consistent feature at this station.**

Similar to phytoplankton, zooplankton, in particular copepods, may use DMSP for osmoregulation and were shown to increasingly assimilate DMSP at higher salinities (Tang et al., 1999). Likewise, DMSP ingestion by copepods increased with increasing DMSP content of the food (Tang et al., 1999). The observation reported by De Angelis & Lee (1994) showed that *T. longicornis* feeding on DMSP-rich Dinophyceae (*Prorocentrum minimum*) resulted in the highest methane production rates per copepod, which suggests a link between the DMSP content of the diet and zooplankton-associated methane production. Based on the observed zoo- and phytoplankton distributions (Fig. 8, 9 and 10), we speculate that the development of distinct subthermocline methane enrichments in the central Baltic Sea is influenced by the combination of *T. longicornis* and relatively high abundances of Dinophyceae.





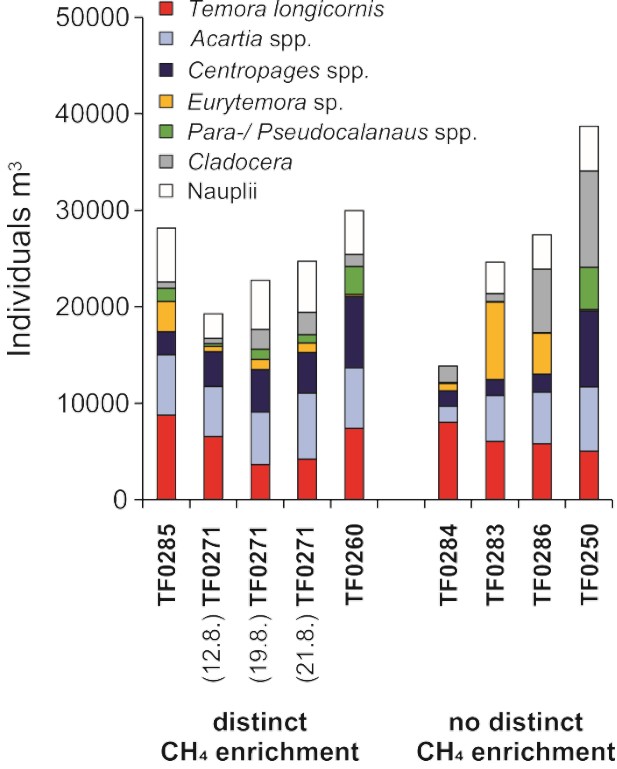

**Figure 10: Average zooplankton community composition between the sea surface and the halocline for stations with distinct (left bars) and without distinct (right bars) subthermocline methane enrichments. Three replicate samples were taken at station TF0271 in a period of ten days to investigate the temporal variability in the community composition at this location.**

To retrieve further information about potential trophic relationships between *T. longicornis* and Dinophyceae lipid biomarkers of concentrated plankton samples from two depths, the surface layer and from subthermocline waters were analyzed. Also, samples from stations with (TF0271) and without (TF0250) distinct subthermocline methane enrichment were compared. Although individual fatty acids have to be assigned cautiously to specific taxonomic groups, the distribution of these compounds in copepod neutral lipids has been shown to largely reflect the lipid composition of the prey (Peters et

al., 2013). Fig. 11a shows the neutral lipids extracted from the surface layer phytoplankton obtained at station TF0271. The fatty acids contained in samples from this depth revealed major contributions from cyanobacteria (typically high in 16:0 and $C_{18}$ fatty acids), diatoms (high in 20:5), and Dinophyceae (22:6, cf. Peters et al., 2013), which corresponds to the phytoplankton community composition observed at this station by microscopy (Fig. 8a). The relatively high abundance of Dinophyceae in the surface layer was further reflected in the presence of 24-norcholesta-5,22-dienol ($26^{5,22}$), an unusual

sterol that is regarded as a specific marker for these algae in temperate waters (Rampen et al., 2007).

Compounds extracted from the subthermocline zooplankton community at station TF0271, dominated by *T. longicornis* (52 %), revealed a broad similarity with the phytoplankton-derived lipids. It also consistently reflected the diurnal feeding behavior of these copepods in the surface mixed layer (Fig. 11b). Notably though, lipids of putative dinophyte origin were



considerably enriched, supporting the idea of a preferential uptake of these algae by *T. longicornis*. In contrast, mixed zooplankton obtained at the reference station TF0250, contained few *T. longicornis* (10 %) but relatively more *Acartia* spp. (37 %) and *Pseudocalanus* spp. (32 %). This sample showed much lower relative amounts of the Dinophyceae-derived biomarkers 22:6ω3 and 26[5,22], indicating only a minor importance of this food source at the reference station (Fig. 11c).

5    Altogether, our biomarker data further corroborates our suggestion that the feeding of *T. longicornis* on (DMSP-rich) Dinophyceae may be an essential factor for the development of subthermocline methane enrichments in the central Baltic Sea.





**Figure 11: GC-MS chromatograms of neutral lipids (methyl ester/TMS derivatives) from mixed layer phytoplankton (a) and subthermocline zooplankton (b) at station TF0271 (with a distinct subthermocline methane enrichment). Data from subthermocline zooplankton at TF0250 (without a distinct subthermocline methane enrichment) are shown as a reference (c). Main compounds are labelled and interpreted as follows (see text for further discussion); 16:0, *n*-hexadecanoic acid (unspecific, high in bacteria); 14OH/16OH, *n*-tetradecanol and *n*-hexadecanol (copepod wax esters); 18:1, oleic acid (unspecific, high heterotrophs but also in (cyano)bacteria); 20:5, *n*-eicosapentaenoic acid (phytoplankton, high in diatoms); Std, internal standard; 22:6, *n*-docosahexaenoic acid (phytoplankton, high in Dinophyceae); $26^{5,22}$, 24-norcholesta-5,22-dienol (specific for Dinophyceae); $27^5$, cholesterol (unspecific, high in zooplankton, but also found in some algae including Dinophyceae). Compounds indicating contributions from Dinophyceae-derived lipids are highlighted with an arrow. Note enhanced levels of these biomarkers in *T. longicornis* - dominated zooplankton at the station with a distinct subthermocline methane enrichment (TF0271).**

We further assume that zooplankton-associated production of fecal pellets and the transit of these pellets through the water column plays a critical role in the build-up of the water column DMSP pool in the central Baltic Sea. Sinking velocities are low for pellets produced from Dinophyceae (Hansen & Bech, 1996; Thor et al., 2003) and we propose that high degradation rates of fecal pellets and the microbial turnover of the contained DMSP (Tang, 2001) to methane contributes to the subthermocline methane enrichment. The gradual loss of DMSP from fecal pellets could plausibly explain the increase of copepod-specific methane production over time, as measured in our incubation study (Fig. 6b and 7). Unfortunately, no Dinophyceae culture was available for our field experiments, because of their relatively low growth rates (Carpenter et al., 1995) and mixotrophic feeding requirements (Tong et al., 2010) that did not allow an adequate radiolabeling of the culture with sodium bicarbonate. Likewise, the relatively low production rates observed in our experiments could be explained by the lack of appropriate substrates for methane production, as Cryptophyceae (i.e. *Rhodomonas* sp.) contain only low amounts of DMSP as compared to other phytoplankton groups (Dong et al., 2013). Furthermore, we used 0.2 µm sterile filtered seawater for our incubations, which should be largely free of microorganisms. However, these microorganisms might be relevant for the turnover of DMSP to methane outside of the copepod bodies. For future studies, we recommend using unfiltered in situ water for these incubations. We further suggest cultivating Dinophyceae under controlled laboratory conditions before the field campaign and feeding these radiolabeled organisms to in situ copepods directly after sampling in the field. These incubations should be accompanied by a quantification of the DMSP and DMS content in phyto- and zooplankton as well as in the water column.

## 4 Conclusion

Several processes that produce methane in oxic waters have been recently identified and it is assumed that climate change will impact their source strength, with far reaching consequences for methane flux and climate feedback. However, mechanisms and magnitudes of this source remain vague. Based on our findings, we suggest that zooplankton contributes to subthermocline methane enrichments in the central Baltic Sea by: (1) direct methane production within the digestive track of copepods and/or (2) indirect contribution to methane production through release of methane precursor substances into the surrounding water, followed by microbial degradation to methane outside the copepod body. Our field observations combined with lipid biomarker studies indicate that a distinct food web segment consisting of DMSP-rich Dinophyceae and

the copepod *T. longicornis* may foster the buildup of methane anomalies in oxic waters of the central Baltic Sea. Recent limnic, terrestrial and marine studies have indicateed that pathways of oxic methane production are in part transferable among these habitats. To obtain a comprehensive understanding of the various controls on oxic methane production, we suggest combining the expertise of these various disciplines in future scientific initiatives.

**5 Data availability**

The data presented here are included as supplemental material.

**Author contribution**

Beate Stawiarski, Oliver Schmale, Stefan Otto designed and built the methane stripping-oxidation line and performed the incubation experiments. Janine Wäge and Natalie Loick-Wilde supported the plankton sampling and the grazing 10 experiments. Volker Thiel and Anna K. Wittenborn conducted the lipid biomaker analysis. Ulf Gräwe performed the oceanographic modelling of the upwelling event. Stefan Schloemer performed the stable carbon isotope analyses. Gregor Rehder, Matthias Labrenz, and Norbert Wasmund helped with the data analyses and interpretation. All authors co-wrote the manuscript.

**Competing interests**

15 The authors declare that they have no conflict of interest

**Acknowledgements**

We thank Nicole C. Power Guerra for her help at sea and in the laboratory. We would also like to thank Michael Glockzin for producing the map and Juha Hatakka (Finnish Meteorological Institute) for providing the atmospheric methane concentration data from station Utö. Further, we appreciate the critical comments by Jörg Dutz. We thank the captain and 20 crew of R/V *Alkor* for technical support. This work was supported by the German Research Foundation (DFG) through grant SCHM 2530/5-1 to O. Schmale, DFG grant LA 1466/10-1 to M. Labrenz, and DFG grant LO 1820/4-1 to N. Loick-Wilde.

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
