# Peer review of "Controls on zooplankton methane production in the central Baltic Sea"

_Biogeosciences, 2018_

## Referee Comment (RC1) · Anonymous Referee #1 · 24 Aug 2018

Comments on 'Controls on zooplankton methane production in the central Baltic Sea' by Stawiarski et al; ms submitted to Biogeosci Discussion; doi: 10.5194/bg-2018-345.

General comments:

The processes which lead to the oceanic methane paradox (i.e. the unexpected accumulation of methane in the oxic surface/subsurface layers of the oceans) are still under debate. Various microbial, biological, and chemical processes have been suggested as possible explanations for the oceanic methane paradox during the last decades. Among these is the suggestion that zooplankton grazing on marine phytoplankton produces significant amounts of CH4. In the ms under review the authors present new results of experiments to decipher the CH4 production by zooplankton in the Baltic Sea. The results presented shed new light on the still enigmatic accumulation of methane in

oxic marine environments. The manuscript is well written and concise and the majority of the conclusions are justified by the presented data. However, I have a few major concerns (see pts 1), 9)-11) below)and, therefore, I can recommend publication only after major revisions.

Specific comments:

1) Introduction: The introduction looks more like a review. A lot of details are given; however, they do not help at all to place the new data/results in a broader context. I suggest to shorten the introduction and to focus it on the main points which are relevant for the discussion of the results.

2) Introduction: The photochemical source of CH4 is not mentioned. This source might be important in the Baltic Sea in view of the fact that the surface layers of the Baltic Sea are heavily influenced by rivers and rich in CDOM, see [Zhang and Xie, 2015].

3) Introduction: DMSO can be a precursor for CH4 as well, see e.g. [Zindler et al., 2013]. This source should be mentioned.

4) Page 3, lines 5/6: The statement that ocean and lakes contribute about 20% to the global natural CH4 emissions is strongly misleading. The Baltic Sea is not a lake, thus, its emissions contribute to the oceanic/coastal CH4 emissions which, in turn, contribute only 1% or less to the global CH4 sources. Oceanic emissions are, therefore, a minor source of atmospheric CH4. In the latest IPCC Report (which is also cited in the manuscript) oceanic emissions are not mentioned as a separate source because of their minor role for the CH4 budget. Please rephrase this statement.

5) P3L14-20: The role of methylphosphonate (MPn) as source of CH4: There are contradicting results: Valle and Karl [del Valle and Karl, 2014] showed that CH4 is not produced from dissolved MPn but rather from particulate/particle-bound MPn (pMPn); which seems to be in contrast to the results by Repeta et al., 2016.

6) P9L30: ' ... since it represents.' I think a part of the sentence is missing.

7) P12L7: Please remove 'sea/air exchange' from the subtitle. The air/sea exchange of CH4 (i.e. CH4 fluxes) is not presented or discussed at all.

8) P12L9/10: please present an equation (incl. a ref for the solubility of CH4) for the calculation of the CH4 saturations. (the equation should be placed to the Method section).

9) P15L5-7: In view of the large standard deviations, the given averages do not seem to be significantly different. Therefore, the conclusion of a composition-depending CH4 production is not justified. Please provide the results of a statistical test which shows that the two averages are significant indeed.

10) P16L4/5: '. . . may have reflected a response of the animals to stress of being removed from their natural environment'. This statement questions all of the presented results. When the zooplankton is that much stressed, how did the authors make sure to get reliable/representative results?

11) P17, discussion. I agree with the line of arguments for DMSP as a potential CH4 precursor. However, there might have been other particle-bound potential CH4 precursors around, e.g. DMSO and MPn. As said above, del Valle and Karl (2014) showed that pMPn is resulting in CH4 production. So, I wonder whether the discussion is only considering DMSP.

12) P17L19: The authors 'believe'? We are scientists, that means we discuss results on the basis of arguments. Please rephrase.

13) P24L23-27: This part of the text should be moved to the Conclusion section.

References:

del Valle, D. A., and D. M. Karl (2014), Aerobic production of methane from dissolved water-column methylphosphonate and sinking particles in the North Pacific Subtropical Gyre, Aquatic Microbial Ecology, 73(2), 93-105.

Zhang, Y., and H. Xie (2015), Photomineralization and photomethanification of dissolved organic matter in Saguenay River surface water, Biogeosciences, 12(22), 6823-6836.

Zindler, C., A. Bracher, C. A. Marandino, B. Taylor, E. Torrecilla, A. Kock, and H. W. Bange (2013), Sulphur compounds, methane and phytoplankton: Interactions along a north-south transit in the western Pacific Ocean Biogeosciences, 10, 3297–3311.

---

## Referee Comment (RC2) · Anonymous Referee #2 · 28 Aug 2018

Stawiarski et al. present measurements and experimental data, which they obtained to identify the origin of elevated methane concentrations in the oxic subsurface water of the Baltic Sea. They tested a hypothesis forwarded by Schmale et al. (2018), if spatial heterogeneity of subsurface methane concentrations result from differences in the copepod community and the associated food web. Therefore, they sampled zooplankton and phytoplankton. They incubated zooplankton communities of different surface and subsurface waters, fed them specific phytoplankton, and derive the methane production rates. Their experiments show that fed zooplankton generate methane and additional methane might be generated by microbial turnover of DMSP.

Overall, it is a well-designed study and manuscript. I found it very interesting to read and have only a few comments.

The spatial heterogeneity of the subsurface methane peak is partly discussed as the result of upwelling (page 14). In figure 4, the authors show the development of the surface water temperature indicating upwelling along the Swedish coast and southeast of Gotland. They relate the high methane concentrations in the surface water at station TF0284 to the upwelling, but do not discuss the other stations. If I consider the positions of the stations, upwelling might have influenced the waters at station TF0283, too. Therefore, I suggest 1) to add the locations of the other stations to figure 4 (longitudes and latitudes are missing on the maps, too) to identify where upwelling might have affected the subsurface methane concentration and 2) consider the increased mixing to modify the plankton community. Therefore, the spatial heterogeneity might be mainly due to upwelling and not a consequence of different plankton communities.

The calculation of the specific activity of the phytoplankton (page 9) includes three terms that are constants, thus, can be ignored and only the first term, i.e. disintegration filter, which is the activity of the phytoplankton, should be presented in table 1. The three constants are the constant for converting DPM to MBq, the added activity, and the specific activity of the tracer. Besides, how was the constant for converting DMP to MBq derived? It appears as if a quench correction is included, but usually, the quench of each sample is slightly different and is corrected by applying a quench curve. How was the quench determined? Furthermore, how did you derive the specific activity of the tracer (SAtracer)? Did you measured it or did you use the value provided by the manufacturer? In my experience, the latter is erroneous and should be validated by own measurements. Anyway, the activity of the phytoplankton is needed to calculate the methane production rate, but an equation of the rate is not included in the manuscript. Therefore, please review equation 1 and include an equation of the methane production rate.

Are there any more accurate numbers for the percentages of Cryptophyceae and N. spumigena (page 10)? It is a very broad statement to distinguish between minor and dominant percentage to the total phytoplankton.

[Figure]

Please include if the incubations were done in the dark or not (page 10).

The difference between the measured methane production rates and the ones reported by de Angelis and Lee (1994) (page 16) might not be due to stressed copepods, but might result from filtration of the seawater (page 25). De Angelis and Lee (1994) most likely set up a similar experiment; therefore, their copepods were similarly stressed. Nevertheless, did they use filtered or unfiltered seawater? Could this explain the difference?

Typos: P2L6: space missing between $\pm$ and 6.1 P6L26: space missing between 1.5 and L P6L29: Did Schmale et al. (2018) study the migration of copepod T. longicornis as suggested by citing the paper? Table 1: It appears as if the duration of experiment 1 with surface zooplankton community is missing. P10L3: space missing between Fig. and 8 P12L21: subthermocline, the r is missing

---

## Author Comment (AC1) · 16 Oct 2018

**Replies to Comments on "Controls on zooplankton methane production in the central Baltic Sea" by Stawiarski et al., manuscript bg-2018-345**

We sincerely thank both reviewers for their insightful comments on our manuscript, which have greatly helped to clarify our findings. The main changes made to our manuscript include:

- a shorter introduction which focuses on the main information which is essential for understanding the topic.

- discussing organic sulphur compounds generally as potential $CH_4$ source, e.g. inclusion of DMSO besides DMS and DMSP as methane precursors.

- inclusion of the locations of the stations in figure 6 for considering upwelling as being influential on the plankton community composition

- inclusion of an equation for calculating methane production rates

- discussion about the minimization of stress factors for the physiological response of the animals

Please find below the *original comments* (in italics) along with our replies (in standard).

On behalf of our co-authors with best regards from Rostock,
Oliver Schmale and Beate Stawiarski

**Referee #1**

**General comments:**

*[...] The results presented shed new light on the still enigmatic accumulation of methane in oxic marine environments. The manuscript is well written and concise and the majority of the conclusions are justified by the presented data. However, I have a few major concerns (see pts 1), 9)-11) below)and, therefore, I can recommend publication only after major revisions.*

We would like to thank the reviewer for acknowledging the value of our work and hope that the changes, which were applied will help to clarify the concerns.

**Specific comments:**
*1) Introduction: The introduction looks more like a review. A lot of details are given; however, they do not help at all to place the new data/results in a broader context. I suggest to shorten the introduction and to focus it on the main points which are relevant for the discussion of the results.*

We agree with the reviewer's suggestion and changed to introduction accordingly (see revised manuscript).

***2) Introduction: The photochemical source of CH4 is not mentioned. This source might be important in the Baltic Sea in view of the fact that the surface layers of the Baltic Sea are heavily influenced by rivers and rich in CDOM, see [Zhang and Xie, 2015].***

We agree with the reviewer's comment and added the photochemical source as follows to the list of methane production pathways:

"(v) photochemical production of methane from colored dissolved organic matter (chromophoric dissolved organic matter, Zhang and Xie, 2015)"

The importance of CDOM in the Baltic Sea as a methane precursor might be interesting, but is not subject of this study, that focuses on the zooplankton associated methane production in the central Baltic Sea, where an immediate influence by river plumes cannot be expected. We agree with the reviewer that future investigations in the Baltic Sea are needed to study the importance of CDOM as a possible methane precursor in river plumes.

***3) Introduction: DMSO can be a precursor for CH4 as well, see e.g. [Zindler et al., 2013]. This source should be mentioned.***

We thank the reviewer for making us aware of this additional precursor.

We added DMSO to the Introduction as follows:
"(iv) $CH_4$ production through microbial degradation of dimethylsulfide (DMS), dimethylsulfoniopropionate (DMSP) and dimethylsulfoxide (DMSO) (Damm et al., 2010, Zindler et al. 2013)."

We also added the following statement to the discussion:
"Dinophyceae, in particular the mixotrophic Dinophysis norvegica, were more abundant at stations with a distinct subthermocline methane enrichment. Dinophyceae produce relatively high amounts of DMSP and DMSO compared to the other phytoplankton species observed within our study (Keller et al. 1989, Hatton and Wilson 2007, Caruana & Malin 2014). Also, positive correlations have previously been observed between DMSP and $CH_4$ and DMSO and $CH_4$ in the surface ocean (Zindler et al, 2013)."

***4) Page 3, lines 5/6: The statement that ocean and lakes contribute about 20% to the global natural CH4 emissions is strongly misleading. The Baltic Sea is not a lake, thus, its emissions contribute to the oceanic/coastal CH4 emissions which, in turn, contribute only 1% or less to the global CH4 sources. Oceanic emissions are, therefore, a minor source of atmospheric CH4. In the latest IPCC Report (which is also cited in the manuscript) oceanic emissions are not mentioned as a separate source.***

We agree with the reviewer and deleted the statement from the manuscript.

***5) P3L14-20: The role of methylphosphonate (MPn) as source of CH4: There are contradicting results: Valle and Karl [del Valle and Karl, 2014] showed that CH4 is not produced from dissolved MPn but rather from particulate/particle-bound MPn (pMPn); which seems to be in contrast to the results by Repeta et al., 2016.***

We shortened the introduction to include information, which is essential for the discussion of the presented data. In this way, we reformulated our statement to be more general and included further references

"(ii) bacterial break-down of methylphosphonate (MPn) under phosphate-stressed conditions (Repeta et al., 2016; Karl et al., 2008; Wang et al., 2017; Teikari et al., 2018)"

**6) P9L30: '… since it represents.' I think a part of the sentence is missing.**

This is correct. We deleted the sentence from the Introduction of the manuscript.

**7) P12L7: Please remove 'sea/air exchange' from the subtitle. The air/sea exchange of CH4 (i.e. CH4 fluxes) is not presented or discussed at all.**

We changed the title to "3.1 Subthermocline methane distribution"

**8) P12L9/10: please present an equation (incl. a ref for the solubility of CH4) for the calculation of the CH4 saturations. (the equation should be placed to the Method section).**

We have added the equation in section "2.1. Hydrographical and chemical characteristics of the water column".

Surface water methane saturation is calculated following Eq. (1), where SV is the saturation value, Cw is the measured concentration of methane in seawater and Cequi the concentration in equilibrium with the atmosphere using the solubility coefficient given by Wiesenburg et al. (1979).

$$SV[\%] = \frac{C_w}{C_{equi}} * 100 \tag{1}$$

**9) P15L5-7: In view of the large standard deviations, the given averages do not seem to be significantly different. Therefore, the conclusion of a composition-depending CH4 production is not justified. Please provide the results of a statistical test which shows that the two averages are significant indeed.**

The reviewer noticed correctly that the averages are not statistically significant. We mentioned that the rates were "higher", but did not present any statistics. For clarity we changed the paragraph as follows:

"The incubations with a high proportion of T. longicornis had higher production rates than the Acartia spp. dominated setups (125 ±49 vs. 84 ±19 fmol CH4 copepod-1 d-1). This indicates that methane production may depended on the composition of the zooplankton community (Fig. 6a). However, the differences were not significant (Kruskall Wallis Test (p=0.150, df=1), which may be a consequence of the limited number of incubations."

**10) P16L4/5: '… may have reflected a response of the animals to stress of being removed from their natural environment'. This statement questions all of the presented results. When the zooplankton is that much stressed, how did the authors make sure to get reliable/representative results?**

We are aware that experiments with natural communities may cause stress to the animals. This general problem needs to be addressed by any researcher who conducts similar physiological experiments. Within our experiments we tried to avoid capture and food stress, which were identified to be influential. For clarity we included the following paragraph:

"To lower the capture stress we selected a rather gentle method for sampling and we avoided food shortage, which was shown to be more influential on the decrease in physiological rates (Ikeda and Skjoldal, 1980). Also, we used a food source which was previously shown to be of good quality (e.g. Knuckey et al. 2005 , Koski and Breteler 2003)."

However, we are aware that the rates obtained in our artificial experiments can never be transferred directly into the natural environment but can serve as an essential contribution to study the factors controlling the shallow water methane production.

**11) P17, discussion. I agree with the line of arguments for DMSP as a potential CH4 precursor. However, there might have been other particle-bound potential CH4 precursors around, e.g. DMSO and MPn. As said above, del Valle and Karl (2014) showed that pMPn is resulting in CH4 production. So, I wonder whether the discussion is only considering DMSP.**

We agree with the reviewer's concerns and adjusted the title of the discussion to include different organic sulfur compounds and included DMSO as a possible methane precursor in the discussion section:

"Organic sulfur compounds as possible substrates for methane production in oxic waters"

"They (Dinophyceae) produce relatively high amounts of DMSP and DMSO compared to the other phytoplankton species observed within our study (Keller et al. 1989, Hatton and Wilson 2007, Caruana & Malin 2014). Also, positive correlations have previously been observed between DMSP and CH4 and DMSO and CH4 in the surface ocean (Zindler et al, 2013)."

Teikari at al. (2018) recently found that N. spumigena could produce $CH_4$ when grown on MPn as phosphorus source. Since cyanobacteria are accumulated in the surface water it might be possible that the process described by Teikari et al. (2018) has an impact on methane productionin the surface waters (above the thermocline). This agrees with the methane budget calculation published by Schmale et al. (2018), who point towards an additional shallow methane source, which is required to maintain the measured methane flux into the atmosphere. However, it is difficult to explain how cyanobacteria related $CH_4$ production can substantially contribute to the observed $CH_4$ enrichment below the thermocline, where the biomass of cyanobacteria is comparably low. Furthermore, there are currently no data available about the quantitative distribution of MPn in surface waters in the Baltic Sea. Based on this knowledge gap, we suggest that further studies should investigate the importance of MPn as a possible methane precursor in Baltic Sea surface waters.

**12) P17L19: The authors 'believe'? We are scientists, that means we discuss results on the basis of arguments. Please rephrase.**

We agree with the reviewer's comment and changed the sentence accordingly:

"We therefore suggest that anaerobic methanogenesis by archaea thriving within fecal pellets played only a minor role."

**13) P24L23-27: This part of the text should be moved to the Conclusion section.**

We moved the suggested section to the conclusions. In addition, we shortened the conclusions section to be more concise.

---

## Author Comment (AC2) · 16 Oct 2018

**Replies to Comments on "Controls on zooplankton methane production in the central Baltic Sea" by Stawiarski et al., manuscript bg-2018-345**

We sincerely thank both reviewers for their insightful comments on our manuscript, which have greatly helped to clarify our findings. The main changes made to our manuscript include:

- a shorter introduction which focuses on the main information which is essential for understanding the topic.

- discussing organic sulphur compounds generally as potential $CH_4$ source, e.g. inclusion of DMSO besides DMS and DMSP as methane precursors.

- inclusion of the locations of the stations in figure 6 for considering upwelling as being influential on the plankton community composition

- inclusion of an equation for calculating methane production rates

- discussion about the minimization of stress factors for the physiological response of the animals

Please find below the *original comments* (in italics) along with our replies (in standard).

On behalf of our co-authors with best regards from Rostock,
Oliver Schmale and Beate Stawiarski

**Referee #2**

**General comments:**
*Stawiarski et al. present measurements and experimental data, which they obtained to identify the origin of elevated methane concentrations in the oxic subsurface water of the Baltic Sea. They tested a hypothesis forwarded by Schmale et al. (2018), if spatial heterogeneity of subsurface methane concentrations result from differences in the copepod community and the associated food web. Therefore, they sampled zooplankton and phytoplankton. They incubated zooplankton communities of different surface and subsurface waters, fed them specific phytoplankton, and derive the methane production rates. Their experiments show that fed zooplankton generate methane and additional methane might be generated by microbial turnover of DMSP. Overall, it is a well-designed study and manuscript. I found it very interesting to read and have only a few comments.*

We would like to thank the reviewer for acknowledging the value of our work and hope that the changes, which we applied will address the suggested comments to the reviewer's satisfaction.

**Specific comments:**
*1) The spatial heterogeneity of the subsurface methane peak is partly discussed as the result of upwelling (page 14). In figure 4, the authors show the development of the surface water temperature indicating upwelling along the Swedish coast and*

*southeast of Gotland. They relate the high methane concentrations in the surface water at station TF0284 to the upwelling, but do not discuss the other stations. If I consider the positions of the stations, upwelling might have influenced the waters at station TF0283, too. Therefore, I suggest 1) to add the locations of the other stations to figure 4 (longitudes and latitudes are missing on the maps, too) to identify where upwelling might have affected the subsurface methane concentration. Therefore, the spatial heterogeneity might be mainly due to upwelling and not a consequence of different plankton communities and 2) consider the increased mixing to modify the plankton community.*

We agree with the reviewer's concerns and applied the following changes according to the suggestions:

a) we added the other locations of the other stations to figure 4 and added longitudes and latitudes. These changes helped us to identify that no station was affected by upwelling during our time of sampling. Station TF0283 was indeed at the edge of the upwelling front, but from its temperature depth profile we suggest that the sampled water body must not have been affected, yet. We included the following statement:

"The other stations were not affected by upwelling events during the time of sampling. Even though our oceanographic model output indicated that the water mass at station TF0283 (sampled on the 11th of August) was located at the upwelling front (Fig. 4), our field measurements showed that the station was not affected by the event, as there is no drop in the surface water temperature visible (Fig. 3)."

b) We added the following statement to the introduction:
"Upwelling events can offset water column stratification through a replacement of warm, mostly nutrient-depleted surface water by cooler and usually nutrient-enriched subthermocline waters (Gidhagen, 1987; Lehmann and Myrberg, 2008; Reissmann et al., 2009). Such events may also cause a rapid decline in phytoplankton biomass in the surface water and affect the plankton composition (Vahtera et al. 2005, Nausch et al. 2009, Wasmund et al. 2012)."

c) We also added this statement to the results and discussion:
"However, the phytoplankton biomass was lower at station TF0284, which was recently influenced by an upwelling event. Hence, it needs to be considered that also the phytoplankton composition at this station could have been altered by the event."

*2) The calculation of the specific activity of the phytoplankton (page 9) includes three terms that are constants, thus, can be ignored and only the first term, i.e. disintegration filter, which is the activity of the phytoplankton, should be presented in table 1. The three constants are the constant for converting DPM to MBq, the added activity, and the specific activity of the tracer.*

We agree with the reviewer's comment. However, two out of the three mentioned constants must or may be modified in future experiments according to culture volume and activity of the

tracer. Hence, for better and easier reproducibility we would like to keep the equation in its current form. The present equation (Eq. 2) is also identical with the equation listed in our reference paper published by de Angelis and Lee (1994), who performed similar incubations to measure zooplankton methane production rates.

**3) Besides, how was the constant for converting DMP to MBq derived? It appears as if a quench correction is included, but usually, the quench of each sample is slightly different and is corrected by applying a quench curve. How was the quench determined?**

We used the method described by Jakobs et al. (2013) along with the same equipment and chemicals. Hence we were also able to use the quench curve obtained by those authors.

**4) Furthermore, how did you derive the specific activity of the tracer (SAtracer)? Did you measured it or did you use the value provided by the manufacturer? In my experience, the latter is erroneous and should be validated by own measurements.**

We and other working groups in our institute made a different experience regarding the quality of radiolabels from this company. The quality of the radiolabel (NaHCO3) was confirmed in different studies before and was also checked in the present study. We used the specific activity provided by the manufacturer, which was obtained individually for our solution.

**5) Anyway, the activity of the phytoplankton is needed to calculate the methane production rate, but an equation of the rate is not included in the manuscript. Therefore, please review equation 1 and include an equation of the methane production rate.**

We thank the reviewer for this hint and added an appropriate equation (Eq. 3) to the manuscript.

**6) Are there any more accurate numbers for the percentages of Cryptophyceae and N. spumigena (page 10)? It is a very broad statement to distinguish between minor and dominant percentage to the total phytoplankton.**

To the best of our knowledge no paper that presents any more numbers for the abundance of Cryptophyceae in the central Baltic Sea is available. Based on the IOW monitoring program we calculated the percentage of Cryptophyceae of the total phytoplankton biomass and added the following text:

*"Rhodomonas sp.* may be considered a model representative of the Cryptophyceae, which account for 5.5 % of the total phytoplankton biomass in the Baltic Sea in summer 2016 (IOW monitoring database: https://www.io-warnemuende.de/datenportal.html)."

For the numbers of *N. spumigena* we added the following sentence:

"Here we selected *N. spumigena* as a food source, because this species was the dominant phytoplankton in the surface waters during our field campaign and accounted for 23% of the phytoplankton biomass."

**7) Please include if the incubations were done in the dark or not (page 10).**

The incubations were done in the dark. We included the following statement:

"All bottles were kept in the dark at in situ temperature for 1 to 3 days in temperature controlled incubators."

**8) The difference between the measured methane production rates and the ones reported by de Angelis and Lee (1994) (page 16) might not be due to stressed copepods, but might result from filtration of the seawater (page 25). De Angelis and Lee (1994) most likely set up a similar experiment; therefore, their copepods were similarly stressed. Nevertheless, did they use filtered or unfiltered seawater? Could this explain the difference?**

De Angelis and Lee (1994) used GF/C filtered seawater for their experiments. However, their pore size was slightly bigger (1.2µm) than within our experiments (GF/F, 0.7µm). We included the following statement:

"Another factor which may have led to lower methane production rates than measured by de Angelis and Lee (1994) is the quality of the filtered sea water used in the incubations. In our experiments we used filters with a pore size of 0.7 µm while de Angelis and Lee (1994) used filters with a pore size of 1.2 µm to prepare the incubation water. Our intention was to exclusively investigate the methane production by zooplankton while minimizing the influence of particulate material (e.g. fecal pellets) in the seawater. However, we are aware that the smaller pore size used in our studies may reduced the number of bacteria in the incubation water, which may have been important for the methane production outside the body of the copepods."

**9) Typos: P2L6: space missing between _ and 6.1 P6L26: space missing between 1.5 and L**

We included a space.

**P6L29: Did Schmale et al. (2018) study the migration of copepod T. longicornis as suggested by citing the paper?**

The reviewer noticed correctly that the citation was incorrect. We deleted it.

**Table 1: It appears as if the duration of experiment 1 with surface zooplankton community is missing.**

We moved the duration of the experiment to the center of the cell to be in line with "Exp." and "SAphy"

**P10L3: space missing between Fig. and 8**

We refurmulated this section.

**P12L21: subthermocline, the r is missing**

We included an "r" in "subthermocline".

---

## Author Comment (AC3) · 16 Oct 2018

The comment was uploaded in the form of a supplement:
https://www.biogeosciences-discuss.net/bg-2018-345/bg-2018-345-AC3-supplement.pdf

---

## Author Comment (AC5) · 16 Oct 2018

The comment was uploaded in the form of a supplement:
https://www.biogeosciences-discuss.net/bg-2018-345/bg-2018-345-AC5-supplement.pdf

---

## Author Response (AR2)

Dear Helge, Dear Susan,

**(1) Conversion dpm into MBq:**
we thank you very much for making us aware of our unfortunate mistake that we made during the review process. Indeed, we understand your confusion about the conversion factor from dpm to MBq, which was copied into the new manuscript by us incorrectly. We found that the mistake happened through converting 2,2 x 10^9 dpm/mCi, rather than 2,22 x 10^9 dpm/mCi into dpm/MBq.
Of course the conversion factor of 6 x 10^7 dpm is the appropriate one to use for standard conversion from dpm to MBq! This is now corrected in the manuscript.

**(2) Quench curve:**
We see, that we need to be more specific about the quench correction, though. As we mentioned, we used the same method for quench correction as Jacobs et al. (2013). Their method used the standard quench correction curve provided by the manufacturer for 14C measurements and which was specific for their scintillation cocktail. As they do not mention it specifically in their manuscript we would like to add it to our manuscript. We are aware that quench corrections are generally important when e.g. using natural seawater samples. However, we, and also Jakobs at al. (2013, 2014) do not add any liquid sample to the cocktail, but only flush it with the $^{14}CO_2$ containing gas stream. "*The 14CO2 was trapped in two 10 ml scintillation vials, each filled with 3 ml of a mixture of phenylethylamine and ethylene glycol monomethyl ether (1:7 v/v) (Jakobs et al., 2013)".* Hence the quench should not vary between the samples and we considered the provided quench curve as appropriate for our measurements.
However, for future experiments we will consider using a quench-set to obtain a more accurate quench curve.
We added the following part in the supplement:
"The $^{14}CO_2$ was trapped in two 10 ml scintillation vials, each filled with 3 ml of a mixture of phenylethylamine and ethylene glycol monomethyl ether (1:7 v/v) (Jakobs et al., 2013). 3 ml Ultima Gold MV (Perkin Elmer) scintillation cocktail were added and the samples were stored in the dark for 3 hours before the activity was measured by liquid scintillation counting (Perkin Elmer; Tri-Carb 2800TR). *A quench curve was provided by Perkin Elmer for the corresponding scintillation cocktail.*"

**(3) Additional information to the DURAN bottles:**
We used "DURAN®, borosilicate glass 3.3, clear (GL45)". This is now mentioned in the manuscript.

**(4) New title for section 4**
Reads now "4 Summary and Conclusions"

**(5) Units in figure 7**
8,8887 is now corrected and reads now 0.8887. We also checked the other figures.

[revised manuscript text omitted]